# Nanosecond-Laser Generation of Nanoparticles in Liquids: From Ablation through Bubble Dynamics to Nanoparticle Yield

**DOI:** 10.3390/ma12040562

**Published:** 2019-02-13

**Authors:** Sergey I. Kudryashov, Andrey A. Samokhvalov, Alena A. Nastulyavichus, Irina N. Saraeva, Vladimir Y. Mikhailovskii, Andrey A. Ionin, Vadim P. Veiko

**Affiliations:** 1ITMO University, 197101 St. Petersburg, Russia; samokhvalov.itmo@gmail.com (A.A.S.); veiko@lastech.ifmo.ru (V.P.V.); 2Lebedev Physical Institute, 119991 Moscow, Russia; ganuary_moon@mail.ru (A.A.N.); insar@lebedev.ru (I.N.S.); aion@sci.lebedev.ru (A.A.I.); 3National Research Nuclear University MEPhI (Moscow Engineering Physics Institute), 115409 Moscow, Russia; 4Research Park, St. Petersburg State University, 7/9 Universitetskaya nab., 199034 St. Petersburg, Russia; v.mikhailovskii@spbu.ru

**Keywords:** nanosecond-laser ablation in liquids, plasma, bubbles, high-throughput generation of nanoparticles

## Abstract

A comprehensive picture of the nanosecond-laser generation of colloidal nanoparticles in liquids is nowadays the demand of their high-throughput industrial fabrication for diverse perspective biomedical, material science, and optoelectronic applications. In this study, using silicon as an example, we present a self-consistent experimental visualization and theoretical description of key transient stages during nanosecond-laser generation of colloidal nanoparticles in liquids: plasma-mediated injection of ablated mass into the liquid and driving the vapor bubble, finalized by the colloid appearance in the liquid. The explored fundamental transient stages envision the basic temporal and spatial scales, as well as laser parameter windows, for the demanded high-throughput nanosecond-laser generation of colloidal nanoparticles in liquids.

## 1. Introduction

Laser ablation powered by inexpensive, robust, high-power nanosecond (ns) lasers appears to be a chemically clean, environmentally “green” key enabling process for industrial manufacturing of functional nanomaterials, for example nanoparticles (NPs), in liquids for diverse promising applications:Hydrogen generation, storage, and catalysis [1,2,3];Wound healing, anti-cancer, anti-bacterial, and anti-fouling activity, and biosensing [4,5,6,7,8,9] in biomedicine;Additive nano-assembling of novel materials [10,11,12,13,14];Non-linear nanophotonics [15,16,17];NP-mediated/enhanced nuclear reactions [18] and many others (for the broad reviews, see, e.g., [19,20,21,22,23]).

In the context of these numerous important potential implications of the colloidal nanoparticles, their high-throughput laser fabrication is in high demand to support broader laboratory, clinical, and technological practices employing NPs. Meanwhile, the underlying spatially and temporarily multi-scale fundamental phenomena could be arbitrarily divided into three main transient stages, as seen in Figure 1: (1) nanosecond-scale laser ablation and plasma formation, primarily bringing the plasma-modified ablation products into the contact liquid and driving the initial bubble boost by the plasma pressure [20,24,25,26,27]; (2) formation of nanoparticles during the several bubble expansion and collapse cycles [26,28,29]; and (3) appearance of the colloidal solution and its interaction with the subsequent laser pulses (re-absorption, fragmentation, etc.) in terms of laser power, repetition rates, and scanning velocities [21,30,31]. The more straightforward and practice-oriented stage three was well explored over the last decades [1,2,3,4,5,6,7,8,9,10,11,12,13,14,15,16,17,18,19,20,21,22,23,24,25,26,27,28,29,30,31], based on rather simple optical, electron microscopy, and dynamic light scattering characterization of the resulting colloids across the broad window of the laser and material parameters. Transient NP formation inside the laser-driven vapor bubbles (the intermediate stage two) was envisioned just recently via spatially- and temporally-resolved sophisticated X-ray transmission studies in liquids [26]. However, the reported studies on both stages two and three missed the crucial input information on laser-induced energy deposition, pressure generation, overall mass, and composition of the injected ablation products, which are the main ingredients in understanding and quantitatively modeling these stages. Meanwhile, basic studies on the initial stage one were recently started based on mass-loss control and ultrasonic acquisition of laser ablation/plasma characteristics [31,32].

The main objective of this particular study is, using crystalline silicon as the case material, similar ablation conditions, and a broad number of informative complementary methods, to present the comprehensive—*from A until*
*Ω*—picture of the nanosecond-laser ablation generation of colloidal nanoparticles in liquids. This study is based on enlightening experimental findings and the related supporting theoretical modeling, enabling overall high-throughput laser manufacturing of colloidal nanomaterials on industrial scales.

## 2. Nanosecond-Laser Generation of Nanocolloids: Basic Stages

### 2.1. Nanosecond Laser Plasma: Energy, Pressure, and Mass Input into Bubble

Dry ns-laser ablation is well known as a high-efficiency process in micro-modification, micromachining, generation of nanoparticles, pulsed laser deposition, and laser-induced forward transfer [24,25,26,27,28,29,30,31,32,33,34,35,36]. At the increasing laser intensity, it usually proceeds as:
Surface vaporization of molten materials along (somewhat lower in vapor pressure) their “melt-vapor” binodes, ending up by short condensation of corresponding low-density atomic or small-cluster vapors, in the characteristic ns-laser intensity range ~0.01–0.1 GW/cm^2^ (fluence range ~0.1–10 J/cm^2^ for laser pulse-widths of 10–100 ns) [37,38];Surface phase explosion (homogeneous boiling) in the proximity of their “melt-vapor” spinodes, expelling high-density vapor-droplet mixture with its bimodal size distribution, in the characteristic ns-laser intensity range ~0.1–1 GW/cm^2^ (fluence range ~1–100 J/cm^2^ for laser pulse-widths of 10–100 ns) [31,32,38,39,40]. This event almost coincides with dense-plume facilitated optical breakdown above the ablated surface, the related onset of sub-critical plasma, and ambient shock-wave emission [38,39];The following sub-critical plasma emerges in the characteristic ns-laser intensity range ~1–100 GW/cm^2^ (fluence range ~10–10^4^ J/cm^2^ for laser pulse-widths of 10–100 ns), and regulates/mediates “laser–plasma–target” interactions via a number of basic laser–matter interaction parameters [37]—laser energy coupling (fraction) to plasma,
(1)$$\eta =1.7\times 10^{-6}\frac{\Psi^{9/8}I^{1/2}\tau^{3/4}}{A^{1/4}\mu \lambda^{1/2}}$$
plasma pressure and surface pressurization, mechanical coupling efficiency,
(2)$$P_{a}=0.6\frac{\Psi^{9/16}I^{3/4}}{A^{1/8}\lambda^{1/4}\tau^{1/8}}$$
(3)$$C_{m}=\frac{P_{a}}{I}=0.6\frac{\Psi^{9/16}}{A^{1/8}\lambda^{1/4}\tau^{1/8}}\frac{1}{I^{1/4}}$$
and ablation rate regulated by screening plasma
(4)$$\dot{М}=2.66\times 10^{-6}\frac{\Psi^{9/8}I^{1/2}}{A^{1/4}\lambda^{1/2}\tau^{1/4}}.$$

Here, in these well-established scaling relationships [37] $\Psi =0.5A{\left[Z^{2}\left(Z+1\right)\right]}^{-1/3}$ and I is the laser intensity (W/cm^2^), λ-laser wavelength (cm), τ-laser pulse-width (s), while atomic mass A and average ion charge Z are the material and plasma parameters. In the hot plasma core above the target surface, ablation products during the ns-laser pulse proceed via dissociation/ionization prior to their following recombination/condensation in the adiabatic expansion/cooling stage;
4Finally, deep material melting and superheating by transient bremsstrahlung and recombination plasma emission and its mechanical unloading during plasma adiabatic expansion results in intense, microsecond-delayed expulsion of micro-droplets [39,41,42,43], which is generally not accounted for in Equation (4).


Meanwhile, the relevance of these well-known ns-laser ablation mechanisms in mass removal, driving near-surface vapor bubbles, and in the generation of colloidal nanoparticles under wet ablation conditions was not justified previously. Below, we report a few key experimental findings justifying the claimed relevance of the well-known universal scaling relationships under wet ns-laser ablation, including pressure generation and mass removal as the main pre-requisites for laser generation of colloidal nanoparticles.

Recently, one dedicated study was conducted [31] to enlighten the pressure generation mechanisms under wet ns-laser ablation conditions, resolving the issue whether the universal scaling relationship [37] (Equations (2) and (3)), or that of Fabbro et al. (*P* ∝ *I*^1/2^) [44] is more relevant. In these dedicated experiments, containing very precise, ultrasonic-based focusing onto the stainless steel and copper targets (to avoid pressure generation in water via its optical breakdown [45]), the common scaling given in Equation (3) was confirmed, even though compressive pressure was five-fold higher under the wet ablation conditions. In this work, 1064-nm, 200-ns fiber laser pulses (IPG Photonics) were used to ablate Si wafer under a 2-mm thick water layer as seen in Figure 2a and to measure fast ultrasonic transients by a piezoelectric transducer on the rear side of the wafer as seen in Figure 2b, and the universal scaling relationship in Equation (3) was justified. Specifically, above the phase explosion threshold of 0.4 GW/cm^2^, the normalized compressive pressure *P/I* (mechanical coupling efficiency *C_m_*) raised non-linearly and then, slowly, sub-linearly dropped down at higher laser intensities, following the well-known relationship *C*_m_ ∝ *I*^−0.25^ as seen in Figure 2c with the corresponding pressure values ranging *P* ~ 1–10^2^ GPa [31,32,39,41,42]. This finding indicates that the common universal scaling relationships for plasma-mediated ablation are more relevant for the description of wet ns-laser ablation in its pressure aspect.

Next, the important issue of mass removal during wet ns-laser ablation was explored, using 1064-nm, 120-ns fiber laser pulses (IPG Photonics) to ablate crystalline Si wafer under a 1-mm thick water layer in the plasma-mediated regime, and to measure the related mass-loss per pulse as seen in Figure 3, in correlation with the optical density of the resulting colloidal solutions [31]. The initial ablation range exhibited the super-linear exponent 1.6 ± 0.3, which was also observed later in optical density dependences on *I,* during similar ns-laser ablation of silver- and gold-coated Si wafers [46]. Likewise, ns-laser ablation of silver films with different predetermined thicknesses by 1064-nm, 120-ns fiber laser pulses also demonstrated the almost linear yield of colloidal optical density versus laser intensity [47]. Although such (super)linear trends for mass-loss or resulting colloidal optical density looks quite trivial, one should account for the factor of the effective ablation square ln(I/I_abl_), which is linear for I/I_abl_ ≥ 1, and then saturates only for I/I_abl_ » 1. Upon such correction, the sub-linear mass yield *M* ∝ *I*^1/2^ becomes evident both for the bulk Si and Ag-film ns-laser ablation, justifying the Equation (4) for wet ablation conditions.

Finally, in order to envision plasma-mediated ns-laser ablation under wet conditions, SEM and optical profilometric characterization of single-shot ablation craters produced at different laser intensities in the range of 0.5–2 GW/cm^2^ was undertaken, as seen in Figure 4. These craters demonstrate strong evidence of lateral (surface) and upward (bulk) plasma-driven melt expulsion [39,41,42,43,44,48].

In the single-shot ablation regime, the presence of the ambient water leads to the formation of a foam-like structure of the ablation craters as seen in Figure 4a–e, with a material removal rate as high as 4–5 microns/pulse, as seen in Figure 4f, accompanied by nano/microparticle ejection in this regime as seen in Figure 4g. For comparison, a similar transition from the removal of small nanoparticles to the expulsion of microdroplets was also presented in Figure 3, exhibiting the bulk melt expulsion as a series of microdroplets, similar to the previous observations [41,42]. Moreover, the visible frozen ablation structures are consistent with recent studies [49,50] showing the higher efficiencies of the dry and wet material ablation by ultrashort laser pulses.

### 2.2. Insights into Nanoparticle Generation: Looking from the Bubble Side

Side-view optical imaging (shadowgraphy) in its pump-probe or high-speed recording arrangements [24,25,27,51], small-angle X-ray scattering [26], and time-resolved optical reflection/refraction [52] are the informative methods to characterize transient bubble dynamics. In this work, 200-kHz side-view optical imaging was performed to reveal novel and important features of vapor bubble dynamics in the plasma-mediated ns-laser ablation regime, as seen in Figure 5; Figure 6. First, fast bubble appearance (<10 μs) accompanied the laser plasma plume and eventually grew to the mm-sized dimensions and then shrunk back at t ≈ 150 μs, as seen in Figure 5a,c. The bubble then collapsed, released the counter-jet from the surface [51] as seen in Figure 5 ( t = 170 μs), and mixed up all ablation, dissociation, and condensation species [53] that emerged in the different positions within the bubble, according to the previous small-angle X-ray scattering visualization [26]. This observed process is in contrast with expectations [30,51] that almost all bubble content can reside on the target (crater) surface during the bubble collapses. In the absence of the external active remixing, such microdroplets should contribute, together with colloidal nanoparticles, to the local dynamic colloidal extinction above the ablated spots (“self-limiting” effect) at *N* > 0.5 shot/spot per pass, even though the resulting average colloidal extinction coefficient at the 1-μm wavelength was below 1 cm^−1^, as seen in Figure 3. Obviously, at higher intensities, such colloidal screening should reduce or even saturate ablation rate in the melt expulsion regime, with such saturation not observed in Figure 2 likely owing to the limited intensity range.

After the collapse, at *t* >170 μs the emerging bubble exhibited an asymmetrical shape, apparently following the deteriorated crater topography as seen in Figure 4. The second and third collapses and rebounds appeared much less pronounced, but much faster—with ≈ 50-μs periods against the initial 170-μs period, as seen in Figure 5c. The initial bubble growth stage driven by plasma (P ~ 1–10^2^ GPa [31,32,39,41,42]) exhibited the expansion (mass flow) velocity V_EXP_ of 10^2^ m/s, as seen in the slope of the plot in Figure 5c. This is consistent with the driving pressure P ~ 1–10 GPa [31,32] through the expression
(5)$$V_{EXP}\approx \frac{P}{\rho D(P)}$$
where the shockwave velocity D ~3–5 km/s at P ~1–10 GPa [54], in agreement with the 100-μm thick dark shockwave front emerging around the laser plasma in 10 μs, as seen in the first images in Figure 5a,b and Figure 6.

Furthermore, in the experiments, the second ns-laser pulse arriving at the same position on the Si wafer surface broke the water down exactly in the region where the content of the first bubble was jet-expelled, which resulted in new water-breakdown plasma and much smaller surface plasma, as seen in Figure 6. This finding sheds light on three important phenomena during multi-shot laser exposures: (1) laser energy losses to optical breakdown in colloidal solutions and accompanying plasma screening of the surface; (2) screening size is comparable to the plasma one, so that neither bubble size [30], nor shockwave impacts the size [21]; and (3) dissociation and ionization of the colloidal species by the succeeding laser pulses, coming to the same spot. Moreover, the succeeding merging of the two bubbles for t > 50 μs brings together their very different contents, further complicating the resulting final nanoparticle size distributions.

### 2.3. Pathways to Colloidal Nanoparticles in Solutions

In the context of the initial plasma and subsequent bubble dynamics driven by 1064-nm, 200-ns laser pulses on Si wafer surfaces, nanoparticle generation demonstrates the strong correlation of its optical density and the ablation spot size, as seen in Figure 7a, with the effect influencing not only ablative pressure [38], but also nanoparticle yield [31,46]. This correlation indicates that intrinsically Si nanoparticle yield is intensity-independent in the high-intensity, hot-plasma regime. Meanwhile, at the increasing laser intensity, the peak size of colloidal nanoparticles measured by dynamic light scattering rapidly and monotonously decreases, as seen in Figure 7b, indicating the strong nano- and micro-particle dissociation in the plasma.

Meanwhile, dissociation of ablation products in the transient ns-laser plasma results in a number of other interesting implications: Re-assembling (re-condensation) of ultrafine NPs from atomic/cluster plasma species, as seen in Figure 8;

Assembling of “core-shell” nanoparticles, with the transparent (e.g., nanodiamonds [17] as seen in Figure 9) core particles pre-dispersed inside the ambient liquid and coated by smaller nanoparticles (gold, silver) from the ablative plasmas/plumes;

Chemical modification of re-assembling NPs with solvent molecules via plasma-activated chemical interactions: Oxidation, sulfurization [53], and carbonization [55] (for oxidized and carbonized silicon NPs in water see Reference [9] and Figure 10);

Plasma-activated chemical interactions with dissolved reagents: gold- and silver-capping of ablated Si NPs via plasma-induced and Si-NP surface reduction of HAuCl_4_ and AgNO_3_, as seen in Figure 11. Their plasma-mediated formation mechanism is illustrated in Figure 12, strongly correlating with Figure 3 for bare Si ablation;

Deep alloying and elemental (phase) segregation during re-assembling (see Figure 13 for Au-Si alloyed NPs with elemental segregation [56]).

### 2.4. High-Throughput NP Fabrication: Laser, Scanning and Colloid Collection Regimes

The issue of high throughput production of colloidal nanomaterials raised earlier in the article can now, using silicon as the case material, find a few envisioned recipes:Higher ns-laser intensity provides almost linear wet ablation mass yield ~ng/mJ in the range between the phase explosion and melt ejection thresholds as seen in Figure 3, owing to the intensity-dependent ablation spot size as seen in Figure 7a and justified plasma-mediated mass removal as seen in Equation (4) and Figure 3, resulting in the similar trend for extinction coefficients of the resulting pure silicon, silver- and gold-coated silicon nanoparticles, as seen in Figure 12. The importance of ns-laser plasma in the generation of colloidal nanoparticles was stressed in multiple studies (see, for example, the overview in recent studies [57,58]);Laser scanning should be adjusted, to avoid a nanoparticle-initiated breakdown in bulk colloidal solution as seen in Figure 5 and Figure 6 and related laser energy losses (as well as nanoparticle fragmentation) in one pass through the clear water above the fresh surface spots. Otherwise, ablation efficiency can drop by a factor of 10 (see, for example, Figure 3 for comparison of the theoretical prediction and experimental measurements of the mass loss). Furthermore, some advanced scanning trajectories—e.g., spiral-like—could be realized at high scanning velocities (galvoscanners <10 m/s, polygons <1 km/s [59]) and laser repetition rates ~0.1–10 MHz;Finally, optimized solvent flow and nanoparticle collection should be chosen for long-term production to avoid the aforementioned colloidal breakdown/screening and laser post-modification of the generated nanoparticles.As a result, the advanced semi-industrial milligram per second (~ng × MHz, dry mass) production rates relevant for medicine, pharmacy, material science, and nanophotonics could be scalably achieved [60,61,62], still with sufficient flexibility in laser, scanning, and flow/collection parameters for specific applications.

## 3. Concluding Remarks

In this study, the advanced, comprehensive overview of nanosecond-laser generation of colloidal nanoparticles in liquids is presented for the first time based on versatile multi-scale and multi-method experimental characterization of the basic transient stages during this key enabling technological process. The initial nanosecond laser ablation and sub-critical plasma formation are linked to ablated mass injection into, and pressure exertion onto, the contact liquid by the well-known scaling relationships, now also justified for wet ablation. The following plasma boost of sub-millisecond, sub-millimeter vapor bubble and its consequent collapse drives a jet expulsion of plasma- and bubble-produced colloidal nanoparticles into the bulk liquid. As a result, the repetitive laser exposure of the same spot demonstrates the additional plasma/bubble formation in the ejected colloid above the ablated surface, indicating, in agreement with our laser scanning tests, the optimal spatial inter-shot separation scale for nanosecond-laser high-throughput nanoparticle manufacturing regimes.

## Figures and Tables

**Figure 1 materials-12-00562-f001:**
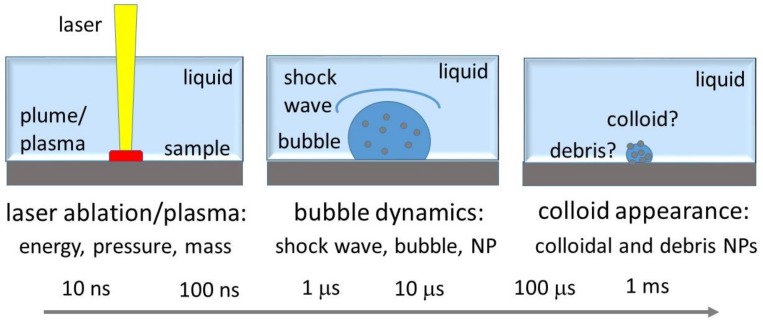
Three basic stages of nanosecond-laser ablation generation of colloidal nanoparticles (NPs) in liquids with their corresponding spatial and temporal scales.

**Figure 2 materials-12-00562-f002:**
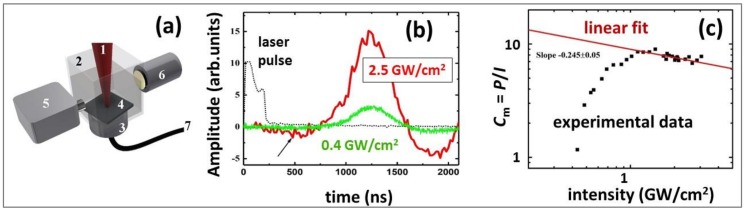
(**a**) Experimental arrangement for ultrasonic characterization: (1) focused fiber ns-laser beam, (2) cuvette filled with liquid (distilled water, thickness of water layer under target ~2 mm), (3) ultrasonic transducer (bandwidth ~30 MHz), (4) silicon wafer, (5) UV-VIS-spectrometer (OceanOptics), (6) light source, (7) to oscilloscope (Tektronix TDS3054C); (**b**) ultrasonic transients at different laser intensities (laser pulse is shown for the reference); (**c**) mechanical coupling efficiency C_m_ versus laser intensity, showing the initial low-intensity range of the threshold-like phase explosion pressure and the succeeding high-intensity range of sub-critical plasma pressure with its linear fit in the double logarithmic coordinates.

**Figure 3 materials-12-00562-f003:**
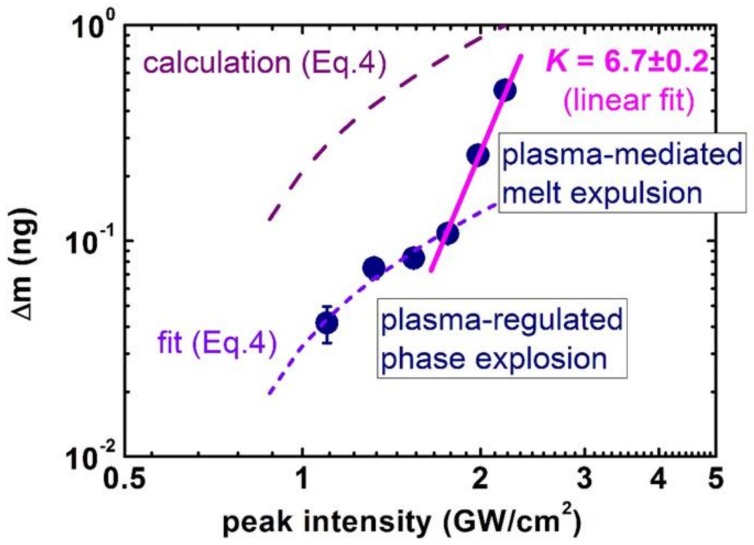
Mass-loss per pulse during plasma-mediated ns-laser ablation of Si wafer in water versus laser intensity (adapted from [31]): Experimental data (circles) and their fits—low-intensity short-dashed curve according to Equation (4) corrected for ln(I/I_abl_) and high-intensity linear one (line). Mass-loss per pulse directly calculated according to Equation (4) is shown for comparison by the top dashed curve.

**Figure 4 materials-12-00562-f004:**
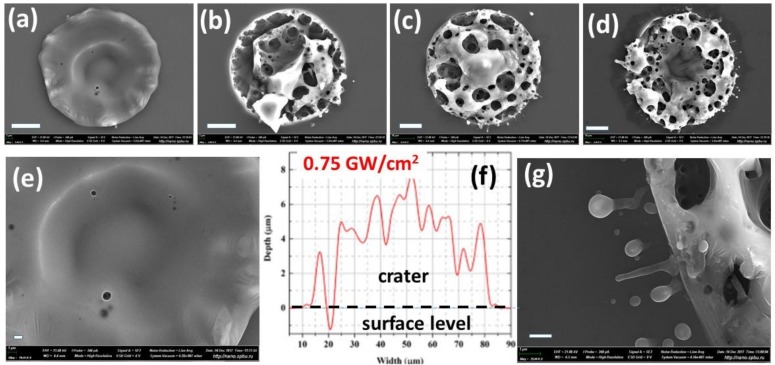
Top-view SEM images of single-shot Si craters in water at different laser intensities of 0.5 (**a**,**e**), 1.0 (**b**), 1.5 (**c**) and 2.0 (**d**,**g**) GW/cm^2^, with the crater profile (**f**) shown for the crater in (**c**). Scale bars-10 μm (**a**–**d**), 1 μm (**e**,**g**).

**Figure 5 materials-12-00562-f005:**
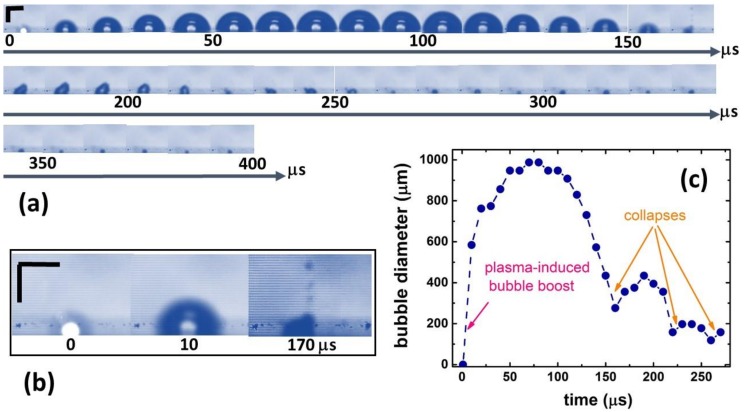
Plasma-driven vapor bubble dynamics on Si wafer surface ablated by single 1064-nm, 200-ns pulses at laser intensity of 4 GW/cm^2^: (**a**) Top three lines: Time series of bubble images (vertical and horizontal scale bars—500 μm), (**b**) Time series of the selected images (vertical and horizontal scale bars—500 μm), (**c**): Dependence of bubble diameter versus time.

**Figure 6 materials-12-00562-f006:**
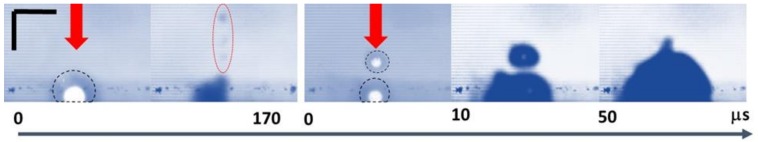
Optical breakdown of water on the Si by the second shot in 10 s (second red vertical arrow along the time scale) at the spatial position of the jet-expelled bubble content, duplication, and merging of the bubbles (vertical and horizontal scale bars—500 μm).

**Figure 7 materials-12-00562-f007:**
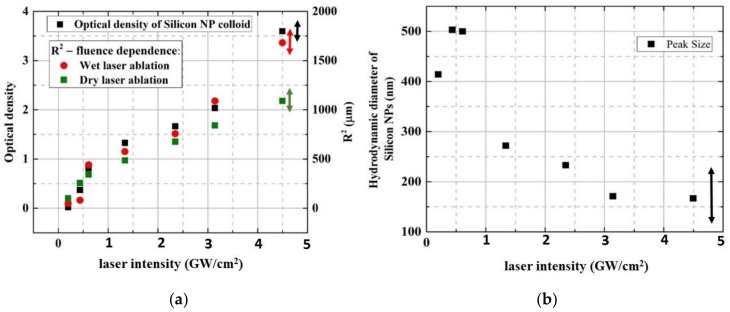
(**a**) Optical density of Si colloid vs. laser intensity in correlation to ablation spot square (for comparison, both dry and wet ablation spot squares are given). (**b**) Hydrodynamic diameter of Si NPs versus laser intensity. The experimental error ranges are shown by the bilateral colored arrows.

**Figure 8 materials-12-00562-f008:**
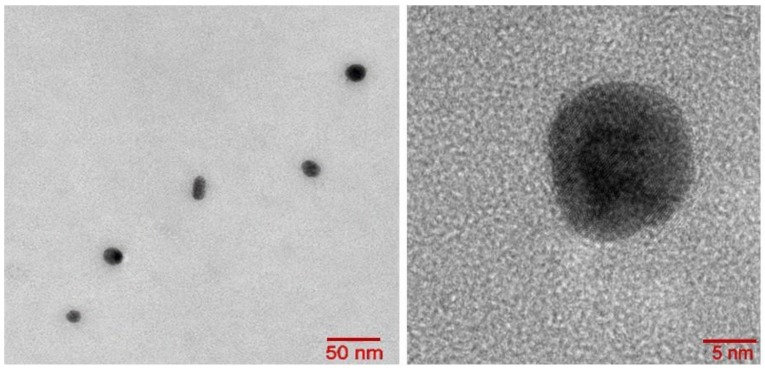
Low- and high magnification TEM images of Si NPs produced during ns-laser ablation of bulk Si in water at the laser intensity of 2.5 GW/cm^2^.

**Figure 9 materials-12-00562-f009:**
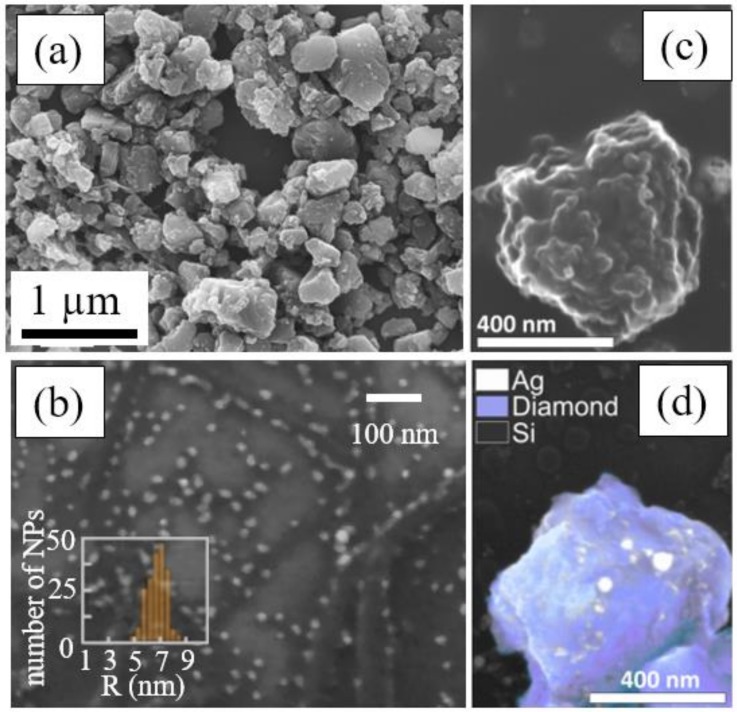
Silver-NP capped nanodiamonds produced via ns-laser ablation of bulk silver sample immersed into nanodiamond suspension in water (adapted from [17]): Top-view SEM images of nanodiamonds (**a**) and silver nanoparticles (**b**), separate nanodiamonds completely (**c**) or partially (**d**) capped by silver nanoparticles.

**Figure 10 materials-12-00562-f010:**
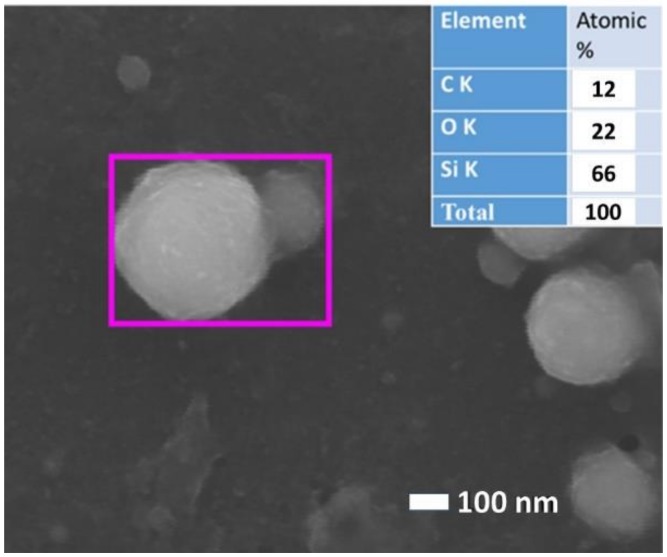
Top-view SEM image of silicon NPs prepared by ns-laser ablation of bulk Si in water. Inset: EDX data in the element content table.

**Figure 11 materials-12-00562-f011:**
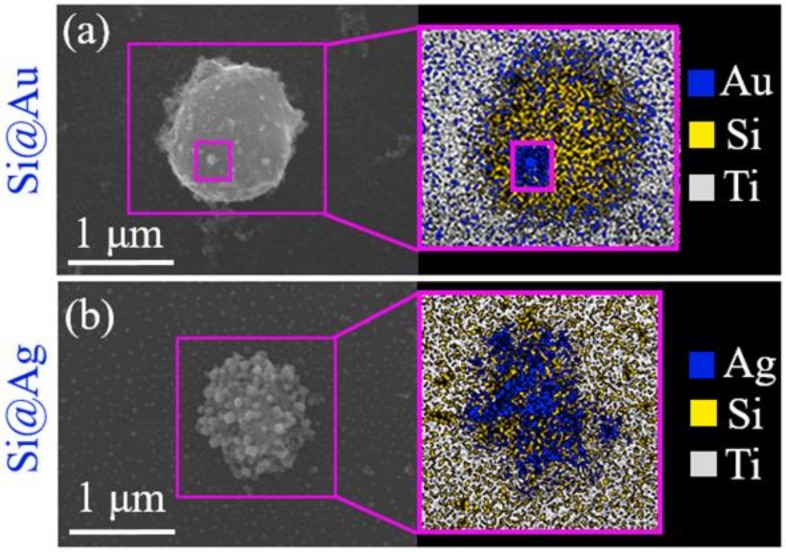
Dry deposits of Au/Si (**a**) and Ag/Si (**b**) NPs on a Si substrate: SEM images (left) and the corresponding elemental analysis by EDX spectroscopy (right) (adapted from [46]).

**Figure 12 materials-12-00562-f012:**
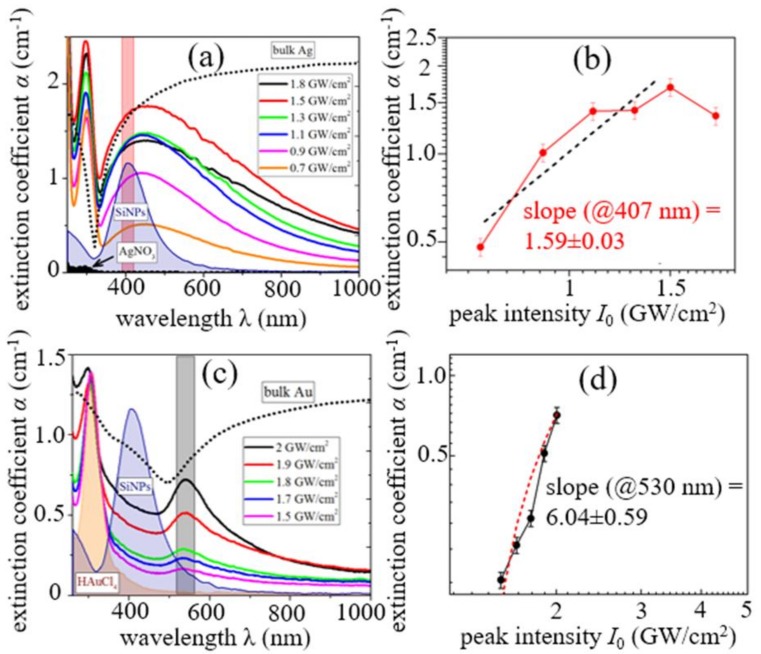
Spectra of extinction coefficients for Si@Ag (**a**) and Si@Au (**c**) colloids fabricated at different laser intensities *I_0_*; (**b**,**d**): The corresponding dependences of the extinction coefficient at 407 nm and 530 nm, respectively (LSPR peak position) on *I_0_* (data presented as log–log plots) with linear approximations.

**Figure 13 materials-12-00562-f013:**
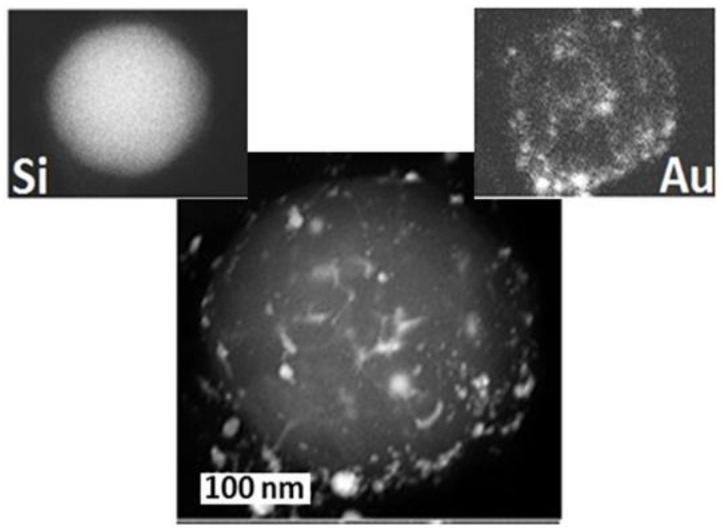
TEM image of Au-Si alloyed NP (Au content ≈5 at. %, see the upper right inset) produced by ns-laser ablation of a thin Au film on a bulk Si substrate in water at the laser intensity ~1 GW/cm^2^, with its Si and Au elemental maps. The scale bar is shown in the bottom image by the frame with the scale mark.

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
