# Peer review of "Nanosecond-Laser Generation of Nanoparticles in Liquids: From Ablation through Bubble Dynamics to Nanoparticle Yield"

_materials, 2019, doi:10.3390/ma12040562_

Reviewer 1 Report

The authors of this manuscript present an interesting approach to present a combined experimental and theoretical description of the nanosecond-based generation of nanoparticles in liquids. A detailed account of results on various temporal scales is derived and presented and this provides a complete description of the underlying physical mechanisms. Results are interesting to a wide researchers focusing on laser-based nanoparticle generation and I would recommend the manuscript for publication in this journal at the present form

Author Response

We agree with the Reviewer.

Reviewer 2 Report

The authors present a complete review of the production of Si colloidal particals by laser ablation. They address all parts of the process and give detailed references. As such I fnd the manuscript very useful.

Author Response

We agree with the Reviewer.

Reviewer 3 Report

The authors discuss the synthesis of nanoparticles by pulsed laser ablation in liquids. They give a general overview of the synthesis process. The paper is sounds and interesting but it can be improved by considering the suggestions below.

Remarks:

The bibliography should be improved. You have way too much self citations (38%), you should bring the self citations ratio down to 25%. You also forgot to mention reviews papers from Wang, Chrisey...

Where are the error bars on Figure 7?

Where is the scale bar on Figure 8 and 10?

Figure 9 is blurry and should be improved. 

Figure 2 should be presented in a single horizontal row. 

All the techniques to measure or characterize the cavitation bubble should be mention to give more in-depth knowledge to the readers. 

Author Response

The authors discuss the synthesis of nanoparticles by pulsed laser ablation in liquids. They give a general overview of the synthesis process. The paper is sounds and interesting but it can be improved by considering the suggestions below.

Remarks:

The bibliography should be improved. You have way too much self citations (38%), you should bring the self citations ratio down to 25%. You also forgot to mention reviews papers from Wang, Chrisey... These and other reviews, as well as 15 more application-oriented “external” references were added.

Where are the error bars on Figure 7? Added

Where is the scale bar on Figure 8 and 10? Added

Figure 9 is blurry and should be improved. Replaced by clear one

Figure 2 should be presented in a single horizontal row. Changed 

All the techniques to measure or characterize the cavitation bubble should be mention to give more in-depth knowledge to the readers. A few more techniques were added

Reviewer 4 Report

This manuscript submitted by Kudryashov et al. gives a nice overview on nanosecond-laser ablation in liquids. I recommend publishing the paper after a minor revision. My comments are listed in the following:

-       “from ablation till bubble…” “till” sounds wrong in this context, I would rephrase the title

-       Abstract: It should read “A comprehensive picture”…

-       In the Intro you say “high-power fiber lasers” however the references you cite are not all using fiber lasers.

-       I don’t understand what you mean by “cooking” of nanoparticles, I would use a more scientific term

-       p.3, l 105, “one special study was conducted”. I don’t see why this study is “special”. I am sure that for every author their paper is “special”.

-       You write that Fig. 3 is from reference [16] however I couldn’t find any similar figure in that reference. Do you cite the wrong reference?

-       P. 5, l 164: mass removal rate as high as 4-5 microns/per pulse. “Micron” is not a unit of “mass”

-       Fig. 8: Scale bar is missing

-       Fig. 9: I cannot resolve the picture in the middle, is there a histogram?

-       P.11, l.305 “laser scanning should be organized” -> “laser scanning should be adapted”

Author Response

Replies to the Reviewer’s comments (in bold)

This manuscript submitted by Kudryashov et al. gives a nice overview on nanosecond-laser ablation in liquids. I recommend publishing the paper after a minor revision. My comments are listed in the following:

 -       “from ablation till bubble…” “till” sounds wrong in this context, I would rephrase the title Changed

-       Abstract: It should read “A comprehensive picture” Changed

-       In the Intro you say “high-power fiber lasers” however the references you cite are not all using fiber lasers. Changed

-       I don’t understand what you mean by “cooking” of nanoparticles, I would use a more scientific term Changed to “formation”

-       p.3, l 105, “one special study was conducted”. I don’t see why this study is “special”. I am sure that for every author their paper is “special”. Changed to “dedicated”, as it was

-       You write that Fig. 3 is from reference [16] however I couldn’t find any similar figure in that reference. Do you cite the wrong reference? Thank you for the notice. It was correctly mentioned in the text (page 4, l 123), however, not updated in the caption to Fig.3; now changed to correct [17]

-       P. 5, l 164: mass removal rate as high as 4-5 microns/per pulse. “Micron” is not a unit of “mass” Changed to “material removal”

-       Fig. 8: Scale bar is missing Added (it was equal to the scale mark)

-       Fig. 9: I cannot resolve the picture in the middle, is there a histogram? Replaced by clear one

-       P.11, l.305 “laser scanning should be organized” -> “laser scanning should be adapted” Changed to “adjusted”